# Grounded Robotic Action-rule Induction through Language Models (GRAIL)

## Abstract

A significant body of recent work illustrates that two components of autonomous planning agents nearly always require manual pre-specification by human experts: the identification and grounding of action symbols (such as "turn right"), and the generation of PDDL action rules (including rule name, parameters, preconditions, and effects). We present the Grounded Robotic Action-Rule Induction through Language Models (GRAIL) system, which, in addition to automating those two processes, also contributes to the expanding research on PDDL model optimization. In this paper, we show how large language models (LLMs) can be used to cluster the sensorimotor experience of the robot and automatically generate useful symbolic abstractions about the robot's capabilities and environment. This language-grounded abstraction allows the learned domain to be modified and used for planning without additional retraining. We evaluate the approach in a standard maze domain and show results for automated symbol identification and grounding, automated rule generation, simulation-based rule validation, and PDDL model optimization. We also discuss and illustrate the advantages of the hybrid neuro-symbolic GRAIL system over traditional symbolic or purely data-driven approaches to similar tasks.

## 1 Introduction

Symbolic abstractions of continuous planning domains (such as PDDL models for robotics) are often useful for enabling faster planning and important constraints are often more easily expressed in terms of abstract symbols. However, manually specifying the symbolic abstractions in a way that is consistent with the underlying continuous domain is difficult – a problem known as symbol grounding. Learning grounded symbolic abstractions from data is also known to be challenging (Goel, 2022).

In this paper we describe a novel approach to learning grounded symbolic abstractions where we hypothesize that by connecting Large Language Models (LLMs) to the physical world through robot sensor and odometry data, we can generate meaningful abstractions of the continuous domain. This technique represents a departure from state-of-the-art systems that require manual pre-definition of symbols (Chen et al., 2023; Silver et al., 2022), enabling a more intuitive interaction between robots and their operational environments.

Furthermore, GRAIL introduces a strategy for automatically generating PDDL action rules. Unlike recent methods requiring manual pre-definition of rules (Song et al., 2023; Li et al., 2021) or employing LLMs for rule creation based on human-defined natural language domain descriptions, (Liu et al., 2023; Guan et al., 2023), GRAIL's approach is based on the meaningful mappings established between symbols and sensor/actuator data during the symbol grounding phase, resulting in a more "context-aware" set of rules. More specifically, GRAIL uses learned grounded action symbols to construct natural language domain descriptions which are then used in a LLM prompt for generating PDDL domains. We know from Liu et al. (2023) and Guan et al. (2023) that using LLMs for natural language to PDDL translation is a viable technique, but in those works, the natural language domain descriptions are primarily human-defined.

A final component of GRAIL is that as the goals of the agent change, it uses a combination of frequency and sensitivity analysis to rank and enable pruning of PDDL action rules. This allows

planning solutions using only action rules relevant to the task at hand (an example of "relevance reasoning" (Levy, 1994)), alleviating scaling issues common with classical planners.

In summary, we present details about the GRAIL architecture and through experiments conducted in a standard maze domain with a Drake-based (MIT, 2023) MOVO simulation (Kinova, 2023), we address the following questions: 1) Can GRAIL automatically identify appropriate action symbols directly from data and associate these symbols with their real-world meanings? 2) Can GRAIL automatically generate valid PDDL action rules? 3) Can GRAIL optimize, in at least a limited sense, the PDDL model? We present experimental results for automatic symbol identification and grounding, automatic PDDL rule generation, simulation-based rule validation, and PDDL model optimization that demonstrate that natural language-based symbols and symbolic systems built upon them can be derived effectively from data with little *a priori* knowledge or human intervention.

## 2 BACKGROUND AND PROBLEM STATEMENT

In this paper we use only fully observable deterministic planning models that can be represented using PDDL 1.0 (McDermott, 2000) with the STRIPS subset (Fikes & Nilsson, 1971). Throughout, we refer to a "PDDL model" as the combination of a "domain" and a "problem" (or "task"). A PDDL *domain* consists of a name, a set of predicates, and a set of action rules (or operators). Action rules are defined with a name, parameters, preconditions, and effects. For example, maze models have predicates such as "at" (`the robot is at cell ?x`), "facing" (`the robot is facing direction ?dir`), and "adjacent" (`cell ?x is adjacent to cell ?y in direction ?dir`). The maze domains have three action rules: move, turnright, and turnleft. An example of a GRAIL generated domain can be seen in Figure 6.A.

A PDDL *problem* (or *task*) consists of a domain name, a list of objects, an initial state, and a goal state. A ground atom is a predicate and a tuple of objects where the tuple dimension is dependent on the number of parameters defined for the predicate. In our maze problems these are primarily cell statuses such as (blocked cell0101) and adjacencies such as (adjacent cell0101 cell0102 right). A state is described by a collection of ground atoms verified to be true, with the assumption that all unspecified ground atoms are false. A goal is represented by a collection of ground atoms affirmed as true in the desired goal state.

The main advancement provided by GRAIL is the connection between the sensorimotor experience of the robot and its ability to auto generate useful symbolic abstractions about its own environment. Specifically, GRAIL takes as inputs raw robotic sensor and odometry data (see the Experimental Results section for a complete list) and uses this data to learn grounded action symbols such as "turn right", and PDDL domains built upon those symbols (see Figure 6.A). In doing this, GRAIL addresses three current problems in autonomous planning agents. Problem #1: is how to define the action symbols themselves and associate the symbols with real world meanings. For example, the symbol "move forward" might be associated with a positive value of X body-axis velocity. Problem #2: is how to learn the PDDL action rules (or "operators") automatically. Problem #3: is that classical planners tend to treat all elements as essential, leading to large complex search spaces.

GRAIL addresses Problems #1 and #2 by leveraging the background knowledge embedded in LLMs to connect clusters of robot sensor data with symbols representative of the appropriate actions. It then uses these grounded action symbols to construct an LLM prompt for generating PDDL action rules and other domain elements. GRAIL's treatment of Problems #1 and #2 taken together provides a potential solution to the symbol grounding problem formalized by Harnad (1990).

Additionally GRAIL addresses Problem #3 by using frequency and sensitivity analyses to rank and optionally prune the model. We use this method to rank the action rules in the PDDL domain and quantify the impact of removing each rule. GRAIL's treatment of Problems #2 and #3 taken together is an example of relevance reasoning (Levy, 1994) or state abstraction (Knoblock, 1994), and is related to the frame problem (Dennett, 1987) in that it deals with understanding what subset of environmental knowledge is relevant to the task at hand.

## 3 THE GRAIL SYSTEM ARCHITECTURE

Figure 1 illustrates each of the GRAIL's four primary subsystems. These subsystems are summarized here in section III and described in more detail in sections IV-VII.

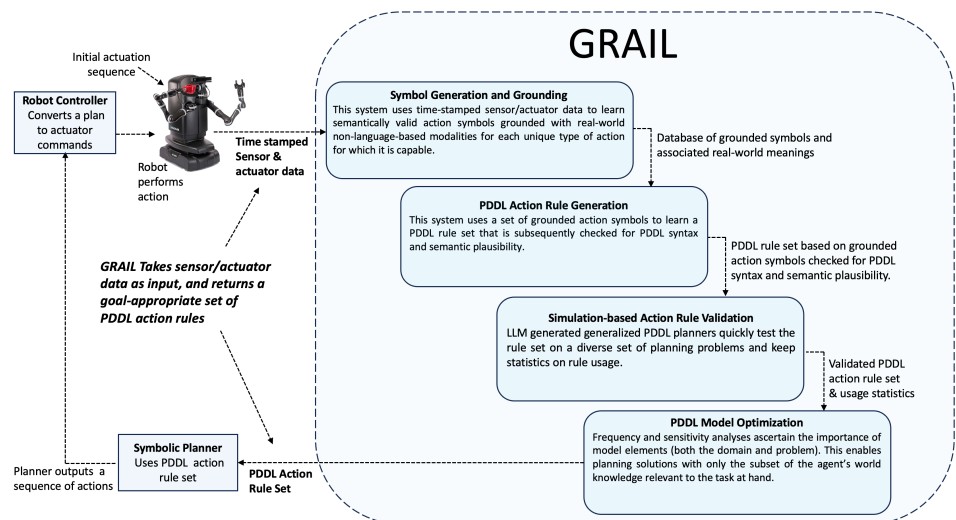

Figure 1: GRAIL System Diagram

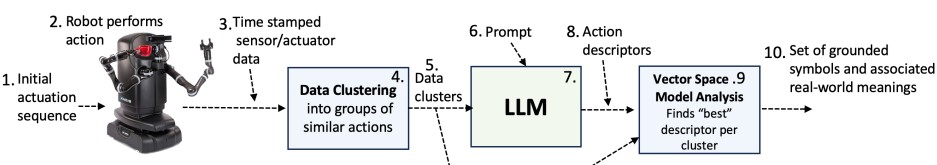

Figure 2: GRAIL Symbol Generation and Grounding Subsystem

*The Symbol Generation and Grounding subsystem* (See Figure 2) automatically generates semantically valid action symbols (such as "turn right" or "move forward"), and grounds those symbols with real world non-language-based modalities. The robot's sensor and odometry data are embedded in prompts to GPT-4 to produce action symbol candidates which are subsequently filtered with a combination of HDBSCAN-based clustering (Malzer & Baum, 2020) and semantic vector space model analysis (Shahmirzadi et al., 2018) to produce a set of grounded action symbols.

*The PDDL Action Rule Generation subsystem* (See Figure 5.C) uses learned grounded action symbols to construct natural language-based descriptions of domains which are subsequently used in a LLM prompt for generating syntactically valid PDDL domains. This is similar to methods that use LLMs for natural language to PDDL translation (Liu et al., 2023; Guan et al., 2023), however, with GRAIL, the natural language domain descriptions are automatically generated.

*The Simulation-based Action Rule Validation subsystem* (See Figure 7) uses PDDL action rules generated by the PDDL Action Rule Generation system as the basis for a PDDL domain. A domain specific, generalized PDDL planner is created based on this domain (Silver et al., 2024). The planner generates plans for a diverse set of planning problems and statistics are kept on rule usage.

*The PDDL Model Optimization subsystem* (See Figure 8) first performs a frequency analysis based on rule usage statistics from the Simulation-based Rule Validation system. This analysis identifies the PDDL rules that are used often (which are kept), never used (which are pruned), or rarely used. A sensitivity analysis (which is task specific with varied domains) is conducted on the rarely used rules allowing these rules to be assigned a value of importance that is based on both frequency of use and sensitivity to removal allowing them to be effectively kept or pruned and enabling tailoring of PDDL domains for specific tasks and goals (such as prioritizing compute cost over planning efficiency).

## 4 SYMBOL GENERATION AND GROUNDING

The GRAIL Symbol Generation and Grounding System (See Figure 1 and Figure 2) enables an agent to automatically learn its own semantically valid action symbols and to ground these symbols with real-world non-language-based modalities. This section describes the complete pipeline, depicted

in Figure 2. The process begins with the generation of sensor and odometry data (Figure 2.1-2.3), which is subsequently grouped into clusters representing similar actions, such as "turning right" (Figure 2.4). This numerical sensor data is embedded in prompts to an LLM (Figure 2.6) to generate candidate action symbols for each observation (Figure 2.8). Finally, vector space model analysis is used to identify the most representative action symbol for each cluster(Figure 2.9). The outcome (Figure 2.10) is a set of action symbols anchored in non-linguistic modalities through the corresponding sensors.

*Numerical Clustering*: The goal of clustering in GRAIL is to group sensor/odometry observations into clusters that represent high level abstract action concepts (such as "turn right"), and not specific values of states (such as $Vx = 0.9$ m/s. Therefore, GRAIL's numerical data clustering (Figure 2.4) attempts to group sensor/odometry data into clusters representative of similar actions. Hierarchical Density Based Spatial Clustering for Applications with Noise (HDBSCAN) (Berba, 2020) was used as the capabilities of HDBSCAN aligned with expected requirements for GRAIL (e.g., we do not want to pre-specify the number/shape of clusters, but we do expect to deal with noise/outliers).

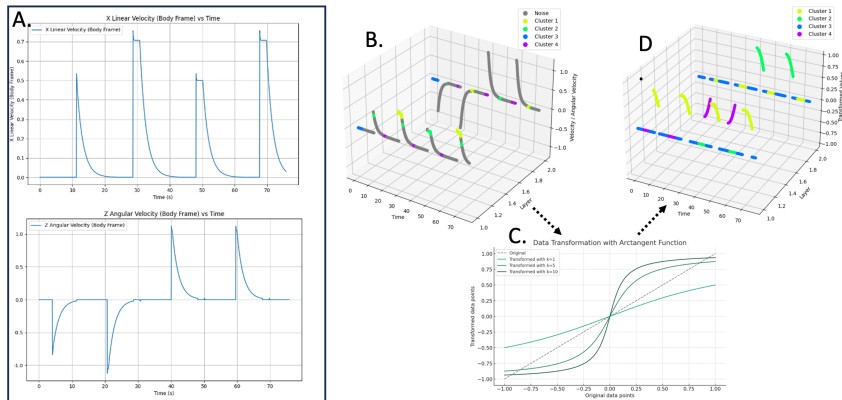

Figure 3: A. Odometry data collected during execution of the initial actuation sequence. X-linear velocity (top) and Z-angular velocity (bottom) B. Raw odometry data poorly clustered with HDBSCAN, C. Scaled arctangent transformation, D. Trasformed data clustered with HDBSCAN.

The GRAIL odometry data, some of which is shown in Figure 3.A, required preprocessing prior to clustering as it has features distinguished by magnitude and HDBSCAN clusters on density. The data are roughly equidistant in Euclidean space regardless of magnitude. In other words, they are roughly uniform in density; so we should not expect HDBSCAN to work well for clustering this data. To illustrate, Figure 3.B is the result of running HDBSCAN on these raw data. Note how most of the points in Figure 3.B end up classified as noise, and the clustering of other points is unintuitive.

Thus, we need to transform this data to a new space in which points with high magnitude in the original space are closer together, and points with lower magnitude in the original space are further apart (increasing the density of the features of interest). We also want a deadband filter to set observations with very small magnitudes equal to zero. To transform the data with these two requirements we applied a scaled arctangent transformation as follows: $f(x) = \frac{2}{\pi} \arctan(k \cdot x)$, where $k = 10$ was chosen by trial and error (this is also shown in Figure 3.C). Applying this transform and then running HDBSCAN with the same parameters used to generate Figure 3.B, results in Figure 3.D. In Figure 3.D, HDBSCAN clusters all points into one of four clusters: Cluster 1 includes points with positive x-linear velocity (moving forward), Cluster 2 includes points with positive z-angular velocity (turning left), Cluster 4 includes points with negative z-angular velocity (turning right), and Cluster 3 is the null action (where the robot is standing still).

*Generating Action Symbol Candidates*: The next step is to generate action symbol candidates for each observation (Figure 2.8). We call them "candidates" as further clustering and filtering will be done to determine from all the "candidates" which is the best descriptor for a given type of action. These candidates are generated by giving an LLM (in this case, GPT-4) a prompt that includes a single observation of numerical data (in this case, a single observation from the data in Figure 3.A). One of the prompts used in the experiments we report is shown in Figure 4.A. Providing the reference frame (a standard right-handed coordinate system with an up-pointing z-axis) had a significant

positive impact on the probability of an accurate action symbol candidate. The LLM then returned a short 1-3 word phrase describing the action of the robot during the instant of that observation.

A.
```
prompt = (
    f"Row {row_number}, Time {values[0]}: Given:"
    f"1) x body axis linear velocity: {values[1]},"
    f"2) z body axis angular velocity: {values[2]}\n\n"
    f"3) Orient with a standard right-handed body-fixed
        coordinate system with an up-pointing z-axis."
    "Give only a short (1-3 word) phrase that describes
        the action of the robot.")
```

B.

| Time Stamp | Cluster # | X-Linear Velocity | Z-Angular Velocity | Action Symbol Candidate |
|---|---|---|---|---|
| 1 | 4 | 0 | -0.8759281211 | turn clockwise |
| 2 | 4 | 0 | -0.8727377684 | turn clockwise |
| 3 | 4 | 0 | -0.8693356676 | turn clockwise |
| 4 | 4 | 0 | -0.8661454682 | turn clockwise |
| 5 | 4 | 0 | -0.8627311549 | "turn clockwise" |

Figure 4: A. The LLM prompt used to generate action symbol candidates from odometry data. B. A snippet of odometry data with associated action symbol candidates

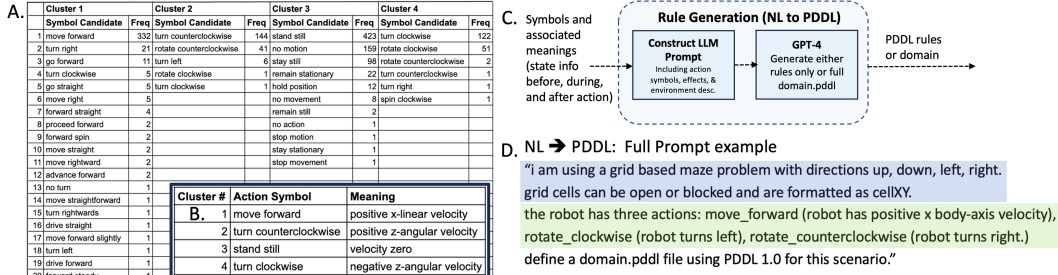

Figure 5: A. Action Symbol Candidates from each of the four clusters along with the number of times each candidate appeared in the data. B. Final action symbols, one from each cluster, chosen with vector space model analysis. C. GRAIL PDDL Action Rule Generation, D. LLM prompt. Green: The part of the prompt derived from GRAIL grounded symbols, Blue: The part of the prompt given as human-provided additional information

We now have an action symbol candidate associated with each data observation. A snippet of this data is seen in Figure 4.B. Figure 5.A shows that the LLM outputs at this stage are mostly semantically correct, but still include noise (misleading or occasionally incorrect action symbol recommendations). Because of this noise we cluster the data into groups of similar actions and then pick the most representative symbol for each cluster (described in the following section).

*Vector Space Model Analysis*: Now that we have our action symbol candidates grouped into clusters of similar actions (for example, rotate left, turn left, and spin counterclockwise might all be in the same cluster), the last step is to find the "best" action symbol to represent each cluster. We do this using vector space model analysis (VSMA) and we tried two different VSMA methods. The first was a frequency only based approach called term frequency-inverse document frequency (TF-IDF) (Sammut & Webb, 2011), and the second used OpenAI's text-embedding-3-small model to create an embedding vector for each action symbol candidate. In both cases, the action symbols were preprocessed to remove all capital letters and all punctuation, then a vector representation for each action symbol candidate was created, the vector centroid of all the vectors was determined, and the nearest neighbor was found. The action symbol candidate associated with this vector then became the final action symbol for that cluster. Thus we end up with a single grounded action symbol for each cluster.

# 5 PDDL Action Rule Generation, Validation, and Optimization

*PDDL Action Rule Generation*: The GRAIL PDDL Action Rule Generation System (See Figure 1) takes in a set of grounded action symbols and automatically learns a PDDL rule set. The method (Figure 5.C) is similar to the methods used by Liu et al. (2023) and Guan et al. (2023) in that it uses an LLM to convert natural language domain descriptions into PDDL files. However, in those works by Liu et al. (2023) and Guan et al. (2023), the natural language descriptions are human-provided whereas here they are constructed using the learned action symbols. Figure 5.D shows an example of such a prompt. This prompt was constructed using grounded action symbols and additional information. The prompt parts that come from action symbols are highlighted in green, and the additional information about the environment and PDDL version (necessary to ensure domain/problem compatibility) is highlighted in blue. This additional information is human provided.

*PDDL Action Rule Validation*: The Action Rule Validation system is shown in Figure 7. The PDDL domain file generated by the Action Rule Generation system along with a PDDL problem file that specified a 10x10 maze (See Figure 6.C) were input to a Fast Downward solver (Helmert, 2011) resulting in a first check on the validity of the rules as Fast Downward will not produce a solution for syntactically invalid rules (See Figure 6.B). For cases where invalid rules were generated, we used an automated interactive debugging scheme as detailed by Silver et al. (2024) and shown in the "Invalid Rule Handling" in Figure 7. We re-prompted the LLM with the errors output by Fast Downward to iteratively refine rules until a valid set was generated (usually in 0-2 iterations). Simulations were then conducted with a Kinova MOVO (Kinova, 2023) simulation in a Drake (MIT, 2023) simulation environment and the robot was able to successfully navigate the maze (See Figure 6.C). Subsequently, we implemented a python-based domain-specific generalized planner as shown in Figure 7 which was capable of quickly and efficiently generating solutions to many different tasks for the same domain. This generalized planner was created following the methodology described by Silver et al. (2024).

*PDDL Model Optimization*: The GRAIL PDDL Model Optimization system (Figure 8) makes use of frequency and sensitivity analyses that can ascertain the relative importance of PDDL model elements (like action rules) in a manner that is, in at least a limited sense, both domain general and problem general. The system makes use of rule usage statistics received from the rule validation system to conduct a frequency analysis of the rules. This gives an understanding of which rules are often used, relatively rarely used, or never used. In addition it uses a domain general sensitivity analysis implemented in Python in a manner similar to the generalized planners in Figure 7 and by Silver et al. (2024). The distinction between the sensitivity analysis and the generalized planner lies in their scope and application. Generalized planners are domain-specific and solve planning problems for a variety of tasks, whereas the sensitivity analysis is task-specific and generates plans using variations of a particular domain. In this case, the sensitivity analysis creates various rule related ablations of the domain by removing each rule in turn and checking the effect on the planning solution cost (which is, in this case, the number of actions in the resulting plan).

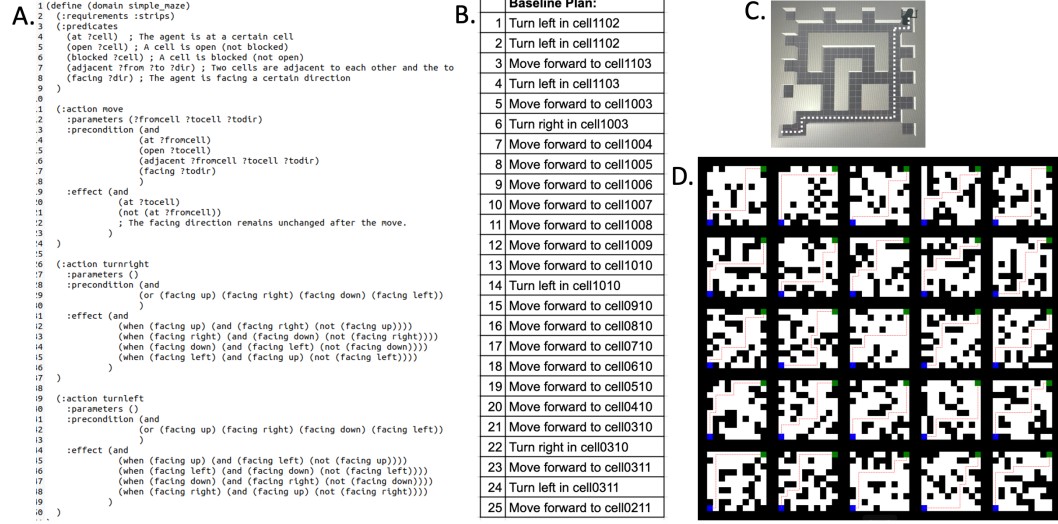

Figure 6: A. GRAIL generated PDDL Domain, B. Baseline solution to the maze planning problem (shown in C.) using GRAIL generated PDDL domain and Fast Downward solveR, C. Drake MOVO maze navigation using GRAIL generated PDDL domain, D. Twenty-five randomly generated 10x10 mazes and solutions generated with a domain-specific LLM-generated generalized planner. Blue cell = initial state, green cell = goal state, red line = solution path

## 6 EXPERIMENTAL RESULTS

We present experimental results for automatic symbol identification and grounding, automatic PDDL rule generation, simulation-based rule validation, and PDDL model optimization that demon-

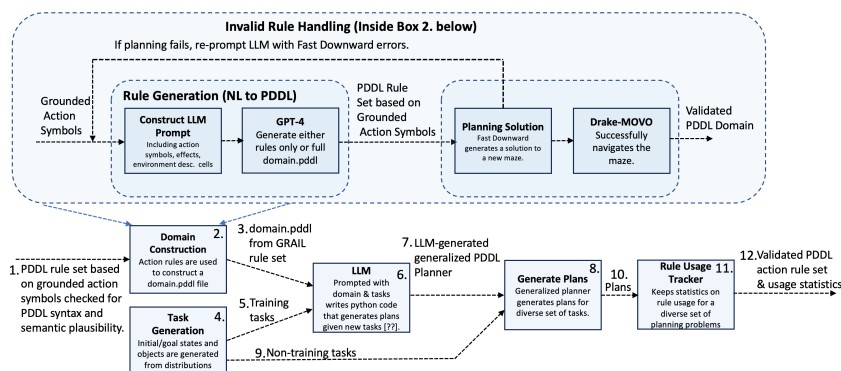

Figure 7: Invalid Rule Handling and PDDL Rule Validation and Usage Tracking

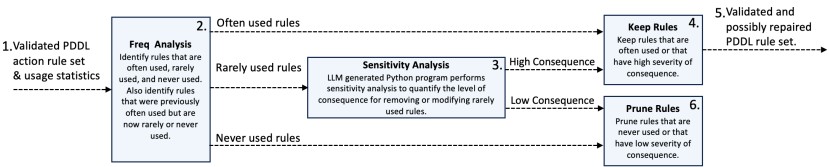

Figure 8: GRAIL PDDL Model Optimization

strate that natural language-based symbols and symbolic systems built upon them can be derived effectively from data with little *a priori* knowledge or human intervention.

*Experimental Setup*: For the experiments we report, a MOVO robot (Kinova, 2023) was simulated in the Drake environment (MIT, 2023) and the initial actuation sequence was chosen to move the robot around an empty room such that it was always either moving forward, turning right, turning left, or standing still (Figure 2.2). These action types, known *a priori* in these experiments, give a clue to the types of actions we should expect to be identified. Robot sensor and odometry data were recorded as the robot executed the actuation sequence (Figure 2.3). Data was collected at 20Hz for 75 seconds resulting in 1,500 total observations. Each observation included 13 values including: $t$ (time stamp), $x, y$, and $z$ (position), $u, v$, and $w$ (linear velocity), $\omega_x, \omega_y$, and $\omega_z$ (angular velocity), pitch angle $\theta$, roll angle $\phi$, and yaw angle $\psi$. Some of this data is shown in Figure 3.A. Other than x-linear velocity ($u$), z-angular velocity ($\omega_z$), and yaw angle ($\psi$), the only other non-zero values for this example were x and y position ($x, y$).

*Symbol Generation and Grounding*: In our maze domain, our clustering process resulted in four clusters. The clusters, action symbol candidates, and frequency are shown in Figure 5.A and the VSMA analysis on these clusters produced the final action symbols and associated meanings shown in Figure 5.B. Both semantic and frequency-based VSMA produced the same results. Notice that the final action symbols are consistent with the action types the robot executed in the initial actuation sequence, and that the associated "meanings" of these symbols are consistent with our understanding of those symbols and associated sensors (i.e., they "make sense").

*PDDL Rule Generation*: A GRAIL-generated PDDL domain is shown in Figure 6.A. While this might look like any number of other PDDL domains for similar tasks, the main takeaway here is that this is an example of a PDDL domain generated automatically from data, with little human intervention beyond the human-provided information in Figure 5.D. This domain was then used used to generate valid planning solutions for the twenty five 2D mazes shown in Figure 6.D.

*Action Rule Validation*: The generalized planner in Figure 7 was used to test the GRAIL-generated domain on the 25 randomly generated 10x10 mazes shown in Figure 6.D. Rule usage statistics were tracked across the task executions and are summarized in Figure 9. Notice that over the course of those 25 mazes, "turn right" was used more than "turn left" (differing from the baseline). As we increase the number of mazes, we expect "turn right" and "turn left" usage to approach equality.

*PDDL Model Optimization*: The PDDL model optimizer (Figure 8) was implemented for the maze problem shown in Figure 6.C. The frequency analysis (in Figure 8) was conducted using statistics only on the baseline case (See Figure 10.A). Results from the frequency and sensitivity analysis

along with rule rankings are shown in Figure 10.B. These results make intuitive sense as the planner can still find a solution (albeit at increased cost) when either the turn right or turn left rules are removed, but removing the move forward rule results in complete failure.

The information generated from the frequency and sensitivity analyses can be used in the process shown in Figure 8 to rank the action rules, and decide whether or not to use all or a subset of them in the actual planning solution. In this process, rules that are often used or that have a high severity of consequence for removal will always be kept. Rules that are never used or that have a low severity of consequence for removal can optionally be pruned.

The results in Figure 10.B can be used to optimize the domain. For example, if our goal is optimal planning efficiency, we know we should keep all three rules; however, if our goal is a valid planning solution with minimal compute cost, we know we can remove either "turn right" or "turn left" and still achieve valid plans, and for this particular maze configuration, "turn right" is the least important, and the best choice for removal (though this may change for different maze configurations).

| | Total | Max | Min | Mean | Standard Deviation |
|---|---|---|---|---|---|
| Move forward | 452 | 20 | 18 | 18.08 | 0.39 |
| Turn left | 68 | 5 | 1 | 2.72 | 1.31 |
| Turn right | 75 | 4 | 1 | 3.00 | 0.89 |
| Total Steps | 595 | 28 | 21 | 23.80 | 1.47 |

Figure 9: Rule usage statistics from the rule validation system for 25 10x10 random mazes. Total is total of times the rule was used, Max and Min are for a single maze

A.
| Scenario | Total Actions | turn right | turn left | move forward |
|---|---|---|---|---|
| Baseline | 25 | 2 | 5 | 18 |
| No "turn right" | 29 | 0 | 11 | 18 |
| No "turn left" | 35 | 17 | 0 | 18 |
| No "move forward | 500 | 241 | 259 | 0 |

B.
| Rank | Rule | Frequency | Weighted Sensitivity | Total Weighted Importance |
|---|---|---|---|---|
| 1 | move forward | 18 | 1900 | 1918 |
| 2 | turn left | 5 | 40 | 45 |
| 3 | turn right | 2 | 16 | 18 |

Figure 10: A. Statistics from the rule statistics tracker (500 = time out/planning failure), B. Frequency and sensitivity results along with final rule rankings.

*Comparing GRAIL with non-symbolic RL-based approaches*: To compare GRAIL with a non-symbolic RL-based approach to the same task, we modified our maze so instead of defining maze cells as "open" or "blocked" we assigned a color to each cell. The initial state was designated as "green" and the goal state "blue". Impassable edge cells were designated as "red", and the rest of the cells in the maze were designated as either white or black. Specifically, this was done by only modifying the human provided information in the LLM prompt used by GRAIL to generate its PDDL domain (See Figure 5.D). This resulted in a GRAIL-generated variation of the domain in Figure 6.A that accounted for these new predicates. We then used this domain with Fast Downward (Helmert, 2011) to enable maze navigation keeping only to the white cells, as shown in Figure 11A.

In parallel, we also trained a non-symbolic Q-Learning-based policy by iteratively updating a Q-table using the following update rule: $Q(s,a) \leftarrow Q(s,a) + \alpha\,(r + \gamma \max_a Q(s',a) - Q(s,a))$, where: $Q(s,a)$ represents the Q-value for taking action $a$ in state $s$, $\alpha$ is the learning rate (set to 0.1 in this experiment), $r$ is the reward received after taking action $a$ in state $s$, $\gamma$ is the discount factor (set to 0.9), and $\max_a Q(s',a)$ represents the maximum Q-value of the next state $s'$. After optimization, this learned policy was also able to successfully navigate the maze keeping only to the white cells (albeit taking a slightly different path) as shown in Figure 11C.

Next, the LLM that generated GRAIL's PDDL domain was prompted as follows: "Give me the same domain with only a single difference. Invert the meaning of black and white". And the LLM produced a modified domain that enabled the agent to traverse the completely unchanged environment, but now keeping only to the black cells as shown in Figure 11B. The Q-learning-based agent was also able to successfully navigate the original maze with the new requirement of keeping to black cells instead of white, but only after retraining that, as shown in Figure 11 D-E, was more costly than the original policy optimization. The increased cost was due to the agent having to overcome the misleading influence of previously beneficial Q-values after initially optimizing its policy for the original environment.

This example illustrates how symbols can be used to represent concepts abstractly. In this case "black" and "white" do not tie directly to specific physical attributes of the environment but instead represent higher-level concepts that can apply under various circumstances. This level of abstraction allows GRAIL to maintain its functionality at a constant cost even when the low-level details (such as whether we want to keep to black or white cells) change. This illustrates one way that

symbolic systems can generalize better across similar tasks because they operate on these high-level abstractions. Whereas, this generalization is more challenging for RL systems that learn policies based on the patterns of rewards and penalties associated with particular low-level states encountered during training. Finally, it is noted that GRAIL realizes the benefits of both symbolic systems, and the "data-driven" benefit of traditional RL systems since both the symbols and the symbolic systems in GRAIL can be generated automatically from data with little *a priori* knowledge or human intervention.

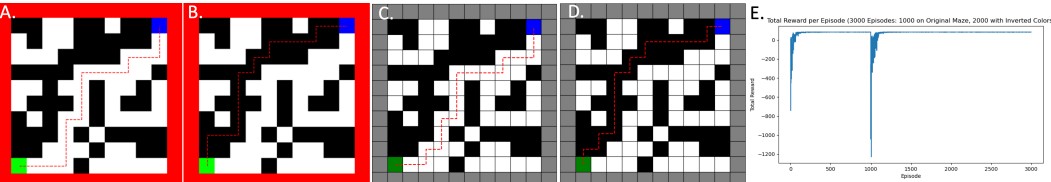

Figure 11: A. GRAIL-based planner navigates the "original" maze. B. After inverting black and white cells, the GRAIL-based planner still successfully navigates through the new environment. C. The learned policy successfully navigates the original maze. D. After inverting black and white, the learned policy succeeds only after re-training, E. Re-training is more costly due to pre-biased policy.

# 7 RELATED AND FUTURE WORK

*Related Work*: It has been well over three decades since Harnad published his seminal work on symbol grounding (Harnad, 1990), and its relevance to autonomous agents has been consistently revisited in the years since (Steels & Belpaeme, 2005; Steels, 2008; Cubek et al., 2015; Rasheed & Amin, 2016). There is a recent renewed focus on the topic as the tasks and operating environments faced by these agents become more complex (Dushkin, 2022; Valenzo et al., 2022).

Concurrent with this re-emerging focus on grounding, Large Language Models (LLMs) have emerged as a transformative force, exhibiting remarkably versatile utility in a wide variety of domains including agent reasoning (Mahowald et al., 2023; Du et al., 2023), mathematical problem solving (Imani et al., 2023; He-Yueya et al., 2023), and robotics (Chen et al., 2023; Song et al., 2023; Driess et al., 2023). However, a well-known limitation of LLMs is their lack of grounding in other real-world non-language modalities.

Many recent efforts employ LLMs in various ways in planning systems. While significant benefits have been realized, these systems have some common limitations. For example, they often require manual pre-definition of symbols (where both the symbols themselves and the meanings of those symbols are provided by human users) (Chen et al., 2023; Silver et al., 2022; Song et al., 2023; Driess et al., 2023) and/or the manual pre-definition of elements of PDDL models (such as the action rules in the domain file) (Chen et al., 2023; Silver et al., 2022; Song et al., 2023; Li et al., 2021). There have also been efforts to use LLMs to directly generate PDDL. Most of these have shared two common limitations (both of which are addressed by GRAIL) 1) the need for natural language descriptions of the domain and/or problem, and 2) lack of grounding. For example, The system developed by Guan et al. (2023) requires natural language inputs as well as other PDDL files and in this system "LLMs may regularly overlook the physical plausibility of actions in certain states."

In addition, several recent efforts attempt to use LLMs to directly solve the planning problem. For example, Silver et al. (2022) use a single PDDL domain file and many PDDL problem files as input to an LLM which then generates Python code that, when presented with other PDDL problem files, can solve a range of problems within the domain. In addition to losing the formal guarantees of classical planners, this system requires a significant amount of context, relies on human-readable names, and performs sub-optimally on domains where certain semantic features or critical aspects of the environment are not explicitly described in the domain files. In contrast GRAIL makes use of and retains the guarantees of a classical planner, and does not require human-generated natural language domain descriptions.

Finally, Chitnis et al. (2022) show transition models akin to PDDL operators can be learned from data. This differs from GRAIL in that the symbols, including predicates for defining preconditions and effects are human-provided, and the system does not generate PDDL action rules or domains.

*Future Work*: The initial actuation sequence for the maze navigation problem was chosen such that the robot was only performing a single type of action at a time (i.e. it was only turning *or* moving forward, but never turning *and* moving forward) and this simplified the data clustering. In the future, we plan to replace the human-defined actuation sequence with an exploration algorithm (possibly similar to the FarMap algorithm (Hwang et al., 2023)) that enables the robot to discover its own action capabilities. In the area of automatic PDDL rule generation future work will include implementation of an improved method that uses a combination of Inductive Logic Programming (ILP) (Cropper & Dumančić, 1991) and Constrained Text Generation LLMs (CTG-LLMs) (Garbacea & Mei, 2022) to induce semantically correct, and syntactically valid, PDDL rules. Finally, we plan to implement GRAIL in more complex experimental domains that incorporate elements of both object manipulation and navigation. In particular, we are interested in experimental benchmark domains that can fail in certain problem configurations, but that can subsequently succeed by learning an additional action rule (and/or predicate).

## 8 Conclusions

This paper presents the GRAIL architecture along with experimental results from each of the four main subsystems shown in Figure 1. The results presented for the symbol generation and grounding system were intuitive and promising. Clustering will become more challenging as the complexity of the initial actuation sequences increase, but the clustering algorithms implemented here should handle this. The GRAIL Automatic Rule Generation implementation (an automatically generated natural language domain description converted to PDDL by GPT-4) worked well. It is yet to be seen how this method handles more complex problems. The GRAIL Rule Validation and PDDL Model Optimization systems also worked as expected, albeit on a simple experimental domain.

The significant contributions of this work include the automation of two processes that have traditionally required manual human intervention: symbol generation and grounding, and PDDL action rule specification. Automation of these processes provides a potential solution to the long standing symbol grounding problem detailed by Harnad (1990). In addition, this work contributes to the current problem of PDDL model optimization. There are multiple efforts underway to rank and prune objects in PDDL problems. Every object that can be removed reduces computational cost, particularly in the operator grounding phase of most classical solvers. GRAIL enables action rules to be ranked and the effect of removing rules to be understood. As GRAIL is intended to operate with an action rule "meta-library" and the rule frequency and sensitivity analyses described in Section VII can be used to choose a subset of that library based on current goals, this is an example of relevance reasoning (Levy, 1994) or state abstraction (Knoblock, 1994). It is related to the frame problem (Dennett, 1987) in that it deals with understanding what subset of environmental knowledge is relevant to the task at hand.

Perhaps the most important potential contribution of GRAIL is the ability for an autonomous agent to explore its own capabilities and environment and generate its own domain description with minimal human interaction.

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
