# OpenReview forum: "Grounded Robotic Action-Rule Induction through Language Models (GRAIL)"
_ICLR.cc/2025/Conference — ICLR 2025 Conference Withdrawn Submission_

### Official Review · Reviewer_1bun · 2024-10-26

**Soundness:** 3
**Presentation:** 2
**Contribution:** 1
**Rating:** 3
**Confidence:** 3

**Summary:**

The paper presents an end-to-end framework for symbolic planning that utilizes LLMs in two different steps. The task is defined as a PDDL formulation and planning problem, with the example throughout of a robot moving to a goal in a grid world with obstacles. The three primary techniques proposed in the paper are 1) using an LLM to assist in converting raw state/action data (i.e. robot velocity) to natural-language symbols, 2) using an LLM with the set of symbols to define the PDDL, and 3) using frequency and sensitivity analysis for identifying parts of the PPDL that could be safely pruned to reduce computational complexity. The paper shows that this framework can be used to generate a valid PDDL, which can be used to plan a path from start to goal in the grid world example, showing potential for LLMs to make the PDDL process more automated.

**Strengths:**

* The problem is well-motivated (clarity)

* The diagrams are clear. I like the labeled letter subsets of the multi-part diagrams. (clarity)

* Applying LLMs as a "classifier" for raw robot motion/sensor data is a fairly creative use of LLM. I applaud the authors for looking to apply LLMs to new parts of the symbolic planning problem. (originality)

**Weaknesses:**

1. I find the novel contribution of the paper to be quite small. While it is true that the LLM can come up with plausible symbol names and a PDDL, there are several major caveats that need to be considered:

  * the input data was heavily tuned before feeding into the LLM: nonlinear data transformation, clustering, and pruning (only some parts of the action were given to the LLM, as seen in Figure 5B. Why was linear/angular velocity in other dimensions not provided?)

  * the output data from the LLM was noisy and required additional hand-tuned clustering, using a non-LLM method. I think an LLM could have been used here

  * The action space used to develop symbols was very small and the actions were simplistic (4 actions total). A more interesting action space, such as using manipulation, or rotation around another axis, would have allowed testing the limits of what the LLM could or could not identify.

  * Some amount of prompt engineering was surely required ("give only a short (1-3 word) phrase", etc.); this in and of itself is expected, but it is yet another piece of human fine-tuning

  All of these caveats greatly reduce the strength of the claim that the identification and grounding of action symbols is "automated" by the system. To strengthen the work, I would suggest that the authors complete more of the pre- and post-processing via more LLM usage, to show that the LLM is capable of providing even more automation.

1. The other potentially-novel part of this paper is that the PDDL is also generated via LLM, and using the previously-learned symbols. However, even this step requires prompt engineering and some symbol information (grid world of cells and a list of directions, even formattting for the cell names) is injected.

1. After these first two contributions, the rest of the pipeline appears to be standard PDDL practice, but the paper continues to explain it in great detail. References to prior work or third-party learning materials on the subject, many of which are already supplied in the paper but also [1], should be sufficient and would help make the contributions of this specific paper clearer.

[1] Aeronautiques, Constructions, Adele Howe, Craig Knoblock, ISI Drew McDermott, Ashwin Ram, Manuela Veloso, Daniel Weld et al. "Pddl| the planning domain definition language." Technical Report, Tech. Rep. (1998).

1. The comparison provided starting on line 404 highlights the strength of symbolic approaches in general, but the only part that is directly relevant to this paper is the sentence on line 419, where the LLM regenerated a modified PDDL. The fast regeneration via natural language should be considered the actual novel contribution in this section, but it is subject to the same drawbacks listed above (still relies on lots of human preprocessing and postprocessing).

1. The use of a dual-armed mobile manipulator is not relevant to the PDDL world or task being performed; this is not a big weakness, but distracts from the fact that this is effectively a continuous gridworld environment. A turtlebot or clearpath robot would do just fine (or alternatively, add manipulation as another action as described above).

**Style Notes***

1. The paper is too wordy, e.g. Fig 1 and section 3, which mostly repeat each other. The last part of Section 2 is nearly word-for-word identical to the end of the second-to-last paragraph in the conclusion. To fix this, Fig 1 and section 3 could be combined, and part of the conclusion could be removed or re-written.

1. There are sentence fragments (beginning with "And") and capitalized words in the middle of sentences.

1. Tables are captioned as Figures.

1. The "most important potential contribution" should be made clearer in the abstract and/or introduction, rather than waiting until the very end of the paper (last paragraph)

**Questions:**

1. In the action clustering step, why was TF-IDF and text pre-processing used here instead of another LLM call? I would expect an LLM to be capable of selecting a single symbol from a list of candidates in a cluster.

1. In Figure 5d (PDDL generation), where does the contents of the parentheses come from? Sentences like "robot has positive x body-axis velocity" do not appear in earlier results.

1. In "PDDL Action Rule Validation" (section 5), is the interactive debugging scheme novel, or exactly as described in Silver et al 2024?

1. Is the Model Optimization step required because the output PDDL from the LLM cannot be trusted to be succinct, or is it generally a nice contribution to the PPDL literature? Is there any prior work in this domain? If so, it is not cited, other than the broad "relevance reasoning" concept. Model optimization is not listed as a "significant contribution" in the conclusion, but then considerable time is spent discussing its benefit there

---

### Official Review · Reviewer_o6d8 · 2024-11-04

**Soundness:** 2
**Presentation:** 3
**Contribution:** 2
**Rating:** 3
**Confidence:** 4

**Summary:**

The paper presents an approach for learning symbolic action labels given execution of actions by a robot in an environment. The approach beings with clustering the odometry data to cluster the execution of the same action and then pass these clusters with expert-provided LLM prompts to VLMs in order to identify action labels. Once these action labels are identified, the approach again uses an LLM to generate a PDDL domain and a problem file and then uses it to plan for a given task. The paper also uses some knowledge pruning techniques  based on frequency of actions used in the validation phase to prune these actions from the generated domain file in order to speed up the planning. Lastly, the approach is evaluated in a single domain consisting of a Kinova robot and 3 discrete actions move_forward, turn_left, and turn_right.

**Strengths:**

The paper targets an important, interesting, and a hard problem. In the current era, where people believe that LLMs are capable of complex reasoning, learning symbolic operators and plannable models are crucial for long-horizon autonomy. While a lot of people have started thinking about these problems, they largely remain unsolved and hard to solve.

The paper is easy to follow. Most of the details that are provided are easy to understand that makes reading the paper easy.

**Weaknesses:**

The paper suffers from some crucial weaknesses and need to be resolved before it can be accepted:

- **Lacks a Lot of Information**: As I mentioned, the paper is extremely clear and easy to read on the information that it provides, however, it lacks a lot of information in general. E.g.,
     - it misses clear information on what are the inputs and assumptions of the system. What kind of actions are known while generating the input trajectories?
     - Symbols mean a lot of things to lot of people. Very late (almost near to experiments) it becomes really clear that what paper is talking about learning is action labels. This needs to be rectified.
    -  Lines 200-209 seems to be arbitrary. No intuition is provided about why this is necessary. The paper should discuss this more naturally.
    -  How are current applicable actions determined for planning? The LLM here is generating PDDL domain meaning also inventing predicates used in the PDDL domain. However, no where in the paper it is mentioned how interpretations of these predicates are determined? How is current state converted into an abstract state and used for planning?
    - One cannot believe that LLM was able to successfully generate correct PDDLs every time. The paper misses the key statistic about the correctness of LLMs as well as details on how the errors were identified and rectified?

- **Unclear Contribution**: The paper is extremely unclear about the contribution. The main contribution claimed are this is the first approach to learn symbolic action labels and other approaches require action labels to be provided as input. Additionally, the approach also claims task planning pruning techniques as one of its contribution.
    - The paper causally (in empirical evaluation, lines 351-352) mentions that action types like move froward and turning right are known apriori and it helps in learning the action labels. If these are already known, how are the action labels learned?
   - A lot of approaches also do not require these action labels and learn them. E.g., Konidaris et al. 2018, Silver et al. 2021, Shah et al. 2024, etc.These approaches don't only learn PDDL actions but also PDDL domains.
    - The task pruning techniques claimed as contribution are already integrate in existing planners such as FF (Hoffman et al. 2001).

- **Extremely Weak Evaluation**: The approach is evaluated in a single setting with a single robot and 3 actions. For the claims what the paper makes, the approach needs to be evaluated in significantly more settings and also compared with existing approaches that learn these actions as well as PDDL models for them.

References:

Hoffmann, Jörg, and Bernhard Nebel. "The FF planning system: Fast plan generation through heuristic search." Journal of Artificial Intelligence Research 14 (2001): 253-302.

Konidaris, George, Leslie Pack Kaelbling, and Tomas Lozano-Perez. "From skills to symbols: Learning symbolic representations for abstract high-level planning." Journal of Artificial Intelligence Research 61 (2018): 215-289.

Silver, Tom, et al. "Learning symbolic operators for task and motion planning." 2021 IEEE/RSJ International Conference on Intelligent Robots and Systems (IROS). IEEE, 2021.

Shah, Naman, et al. "From Reals to Logic and Back: Inventing Symbolic Vocabularies, Actions and Models for Planning from Raw Data." arXiv preprint arXiv:2402.11871 (2024).

**Questions:**

Please refer to the weaknesses mentioned. The questions are interleaved.

---

### Official Review · Reviewer_cVW4 · 2024-11-04

**Soundness:** 2
**Presentation:** 1
**Contribution:** 2
**Rating:** 3
**Confidence:** 5

**Summary:**

This paper presents an LLM-guided method to learn action symbols and operators for PDDL planning. The method produces action symbols by clustering continuous actions and generating descriptions with an LLM. Then it prompts an LLM to generate PDDL domains that define rules of the action symbols, which are further validated and optimized with a PDDL planner and by interacting with the simulated environments. The method is evaluated qualitatively for robot navigation in simulated 2D maze environments against an RL-baed approach.

**Strengths:**

- The paper attempts to tackle the foundamental challenge of learning action and state abstractions (i.e., PDDL domains) for robot planning.
- The proposed method leverages world knowledge in LLM to generate action symbols and operators, which is a promising approach to learn symbols without extensive human intervention.

**Weaknesses:**

- **The method is insufficient to learn grounded PDDL domains.** With the learned action symbols, the method has an LLM to directly generate PDDL domains (e.g., predicates and operators) without human inputs. This doesn't seem to be possible for general PDDL domains because:
  - Learning a self-contained and plannable predicate set is very challenging and usually not possible with an LLM alone. The state-of-the-art methods [1, 2] manage to learn predicates from human feedback and extensive prompt engineering with an LLM. The other work referred in the paper [3] doesn't even learn predicates and operators at all - it assumes known PDDL domain and only translates language goals into PDDL.
  - The operators generated by LLM are not grounded to interaction experience. They should be strictlt aligned with the state change caused by executing actions in the environment [2, 4].

- **The evaluation is insufficient to justify the effectiveness of the method.**
  - The method is only evaluated on a 2D navigation environment that corresponds to a very simple PDDL domain. More comprehensive evaluations on more domains with more complex actions rules is necessary.
  - No quantitative result (e.g., success rate) is reported in comparison with the baseline.


[1] Guan, Lin, et al. "Leveraging pre-trained large language models to construct and utilize world models for model-based task planning." Neurips 2023

[2] Han, Muzhi, et al. "InterPreT: Interactive Predicate Learning from Language Feedback for Generalizable Task Planning." RSS 2024

[3] Liu, Bo, et al. "Llm+ p: Empowering large language models with optimal planning proficiency." ICRA 2023

[4] Silver, Tom, et al. "Learning symbolic operators for task and motion planning." IROS 2021

**Questions:**

- The method has an LLM to directly generate PDDL domains. How are the predicates grounded to sensor observations?
- In experiments, is the RL baseline state-based or image-based?
- What is the quantitative performance of the proposed method and how does it compared to the RL-based baseline?

---

### Official Review · Reviewer_DEdB · 2024-11-08

**Soundness:** 1
**Presentation:** 3
**Contribution:** 1
**Rating:** 3
**Confidence:** 3

**Summary:**

The key contributions of the GRAIL system include:
1. Automation of Symbol Generation and Grounding: GRAIL connects raw sensor and odometry data to grounded action symbols (e.g., "turn right"), addressing the long-standing symbol grounding problem by associating symbols with real-world meanings
2. Automatic PDDL Rule Generation: The system leverages large language models (LLMs) to automatically generate PDDL (Planning Domain Definition Language) action rules based on the grounded action symbols, thus eliminating the need for manual rule specification.
3. PDDL Model Optimization: GRAIL incorporates methods to rank and prune objects in PDDL problems, reducing computational costs and improving the efficiency of classical planners by focusing on relevant elements.
4. Experimental Validation: The paper includes experimental results demonstrating the effectiveness of the GRAIL architecture in a maze domain, showcasing its ability to learn and adapt action rules in response to different problem configurations.

**Strengths:**

1. Automation of Complex Processes: GRAIL automates the traditionally manual processes of symbol generation and PDDL rule specification, significantly reducing the need for human intervention and expertise in defining action symbols and rules
2. Addressing the Symbol Grounding Problem: The paper effectively tackles the symbol grounding problem by establishing meaningful mappings between sensor data and action symbols, which is crucial for enabling robots to operate in dynamic environments
3. The paper is written well and explains concepts and ideas clearly

**Weaknesses:**

1. Expand Experimental Domains: While the paper demonstrates the GRAIL system in a maze domain, the experiments could benefit from being conducted in more complex and varied environments (e.g., dynamic settings with moving obstacles or multi-agent scenarios). This would provide a more comprehensive evaluation of the system's robustness and adaptability. Incorporating real-world tasks, such as navigation in cluttered spaces or interaction with humans, could further validate the system's effectiveness
2. Lack of Diversity of Action Symbols: The paper primarily focuses on a limited set of action symbols. To enhance the system's applicability, it would be beneficial to explore a broader range of action symbols and their corresponding grounding in various contexts.
3. Robustness to Noisy Data: The paper does not address how the GRAIL system performs under conditions of noisy or incomplete sensor data. Implementing robustness tests to evaluate the system's performance in such scenarios would be crucial, as real-world environments often present challenges such as sensor inaccuracies or unexpected changes
4. Comparative Analysis with Existing Systems: While the paper discusses the advantages of GRAIL, a more detailed comparative analysis with existing state-of-the-art systems would strengthen the claims of novelty and effectiveness. This could involve benchmarking against other symbolic and data-driven approaches to highlight specific improvements in performance metrics such as planning time, success rate, and adaptability

**Questions:**

1. How does GRAIL ensure that the grounded action symbols are consistently associated with their real-world meanings across different contexts?
2. The experiments are conducted in a maze domain. How do you plan to extend the evaluation to more complex environments? What are the anticipated challenges in doing so?
3. How does GRAIL perform when faced with noisy or incomplete sensor data? Have you conducted any experiments to assess the robustness of the system under such conditions?
4. Can you provide a comparative analysis of GRAIL with existing systems that also address symbol grounding and PDDL rule generation? What specific advantages does GRAIL offer over these systems?

---

### Note · Authors · 2024-11-21

I have read and agree with the venue's withdrawal policy on behalf of myself and my co-authors.